# Field and Laboratory Observations on the Biology of *Aceria angustifoliae* with Emphasis on Emergence of Overwintering Mites

**DOI:** 10.3390/insects14070633

**Published:** 2023-07-13

**Authors:** Parisa Lotfollahi, Hosein Mehri-Heyran, Solmaz Azimi, Enrico de Lillo

**Affiliations:** 1Department of Plant Protection, Faculty of Agriculture, Azarbaijan Shahid Madani University, Tabriz 5375171379, Iran; prslotfollahy@yahoo.com (P.L.); h.mehri1374.hm@gmail.com (H.M.-H.); s_azimi2007@yahoo.com (S.A.); 2Dipartimento di Scienze del Suolo, della Pianta e degli Alimenti, University of Bari Aldo Moro, Via Amendola, 165/a, 70126 Bari, Italy

**Keywords:** *Elaeagnus angustifolia*, ecology, temperature-mite interactions, plant-mite phenology, sigmoidal model, sticky-band traps, linear degree day model

## Abstract

**Simple Summary:**

*Aceria angustifoliae* Denizhan et al. (Acari: Eriophyidae) is the most common mite infesting Russian olive, *Elaeagnus angustifolia* L. (Elaeagnaceae), in Iran. In the native areas, the fruits of Russian olive are used in folk medicine, and its plants are common in green spaces and parks. This plant is classified as an invasive alien species in North America and *A. angustifoliae* is considered one of its more promising biological control agents. A study carried out from winter 2017 to spring 2019 clarified some biological aspects of its life strategy by means of field observations (sticky trap bands and direct leaf inspections) and exposing overwintering mites to various constant temperatures in the laboratory; a linear regression of the emergence rates of overwintering mites at constant temperatures was applied in order to calculate the lower developmental threshold.

**Abstract:**

Data on the life strategy of *A. angustifoliae* (population fluctuation in buds and on leaves, emergence and migration to the overwintering sites), as well as its temperature-dependent emergence from overwintering sites at constant temperatures, were determined. The eriophyid mite overwintered into buds and the density of active mites inside them from winter 2017 to spring 2018 was higher than that in winter 2018–spring 2019. In the second half of March 2018 and in winter 2018–spring 2019, the mite density inside the buds decreased gradually with a peak of emergence occurring at the beginning of plant blossoming. Population density on leaves increased in summer, reaching a higher and later peak in 2018, and gradually decreased in autumn with mites migrating to overwintering sites. A lower developmental threshold of 4.5 °C was calculated. About half of the mite population was estimated to emerge from the overwintering sites at an accumulation of degree days ranging, on average, between 85.5 (at 20 °C) and 104.4 (at 10 °C) degree days above the assessed threshold.

## 1. Introduction

The eriophyid mite *Aceria angustifoliae* Denizhan, Monfreda, de Lillo et Çobanoglu (Acari: Eriophyoidea: Eriophyidae) was first collected and described from Van, Turkey, on the Russian olive (RO), *Elaeagnus angustifolia* L. (Rosales: Elaeagnaceae) [1], and was later recorded in Serbia, Iran, Uzbekistan, and China [2]. This mite induces leaf distortions along the mid-rib and deep or superficial invaginations of the leaf lamina, which is folded showing a rough surface. The scurfy hairs on the leaf appear longer and more abundant on the deformed surfaces than on the healthy ones, probably due to the lamina distortion or size reduction [1]. Plants with such symptoms have been observed in the northeast and northwest of Iran and have started to show up in other Iranian geographical areas (PL, personal observations). 

In the native areas of this plant species, like Iran, the fruits of RO are used in folk medicine and its plants are common in green spaces and parks, especially in East Azerbaijan province, because of their high tolerance to water stress. On the contrary, RO is classified as an invasive and infesting alien species [3] in North America. *Aceria angustifoliae* is considered its promising biological control agent and its planned field release was recently approved by the Canadian Food Inspection Agency [4].

In general, adult females of eriophyoid mites overwinter mainly under bud scales and bark crevices, where they often aggregate in large numbers. Also, females of *A. angustifoliae* overwinter in buds but data, such as the end of overwintering time and winter–spring emergence, are not available on the biology of this species. Understanding the life history strategy and population dynamics of this plant feeder will allow the setting up and development of an effective management program for the pest and hopefully it can be more efficiently applied as a biological control agent [5]. Consequently, the control of the pest at the emergent overwintering female stage early in the growing season will be crucial to understand [6]. The timing of possible sprays close to the peak period of the spring emergence of mites from buds should allow for a more efficient and less environmentally impacted approach than mid-season control measures, as verified, for example, against the pear rust mite, *Epitrimerus pyri* (Nalepa), and the hazelnut big bud mite, *Phytoptus avellanae* Nalepa [7,8,9].

In addition, considering that the phenology of mites is significantly affected by temperature, the study of the effect of the temperature on *A. angustifoliae* appears to be fundamental for obtaining information on mite development and survival [9]. Many models have been developed in entomology for simulating and forecasting pest biology and applying control strategies based on the climatic trends. This is the case of many phenological models which use growing degree days (DD) for predicting the time for a higher probability that a specific instar/stage of the pest could be found or for how long generations last [10]. In acarology, apart from ticks and other non-plant feeding taxa, these models have been used for some spider mites [11,12,13,14] and eriophyoids [8,15,16] and a few others among the obligate phytophagous mites. They represent a valid and useful tool if well-developed and validated in the field. 

This research was aimed at investigating some biological aspects of *A. angustifoliae* in the field and laboratory with a particular interest in: (i) mite life strategy (mite population density in the overwintering sites; mite migration from and to overwintering sites; mite population dynamics); (ii) development and survival at constant temperatures; (iii) development threshold; (iv) validation of a linear degree day model in order to predict the emergence of mites from overwintering sites.

## 2. Materials and Methods

Field observations and samplings were carried out on RO plants in Azarbaijan Shahid Madani University, Tabriz, Iran (GPS: 37°48′54.8″ N, 45°56′39.7″ E, 1308 m above sea level).

Laboratory observations were made at the research laboratory of Azarbaijan Shahid Madani University, Tabriz, Iran.

### 2.1. Field Observations: Population Dynamics of Overwintering Mites

Samplings of buds were carried out five times from winter 2017 to spring 2018 and four times from winter 2018 to spring 2019. 

Samplings were carried out on five randomly determined trees. Four twigs per tree, each twig in one cardinal direction, were cut at each sampling and transferred to the laboratory for study. Therefore, 20 twigs were sampled on each date. Sampled twigs were approximately 15–20 cm long and had about 10–12 buds each. Three buds per twig were taken randomly—one at the tip, one in the middle, and one at the base of the twig. Their bud scales were separated in a Petri dish within water. Only live motile mites released into the water were counted under a Nikon (Tokyo, Japan) SMZ745 stereomicroscope at a magnification of 50× and the average number of mites per bud was calculated.

### 2.2. Field Observations: Phenology of Mite Emergence

Sticky-band traps were applied from 27 February 2018 to capture mites emerging and dispersing from overwintering sites on infested trees, as used by Bergh (1992) [17] for *Epitrimerus pyri*. The sticky-band trap consisted of a 2 cm wide strip of Parafilm M^®^, tightly wrapped and completely surrounding the twig at the top, just under the distal bud. The strip was covered with pure glue (AG glue 166, Greenagrotech, Republic of Korea). Actually, these bands can measure relative mite activity moving mainly from lateral buds to the apical buds and their new stems rich in leaves. This activity can be considered mainly related to the emergence of mites from overwintering sites even though it might also concern the early intra-plant dispersal. Sticky-band traps were applied on four twigs of each of five randomly selected trees. Trapped twigs were approximately 15–20 cm long, each with about 10–12 buds and oriented in one of a cardinal direction (Figure 1A,B). Weekly, the bands were cut with a razor blade, removed from the twigs and laid flat on glass plane slides, so that the glued surface was facing up. Captured mites were counted under a Leitz (Wetzlar, Germany) LABORLUX S microscope at a magnification of 100×. Bands were replaced weekly with new ones until no more mites were trapped (14 April 2018). 

This study was not repeated in 2019. The low population density found in the buds in winter 2018–spring 2019 suggested we avoid repeating these observations. 

The data on the maximum and minimum daily temperatures of the nearest town (6 km away) to the experiment location, i.e., Gogan, East Azerbaijan province (lat. 37°46′51.7″ N, long. 45°54′32.3″ E, 1294 m above sea level), were used in this study to calculate the degree day corresponding to the emergence of the overwintering mites. Average daily temperature was calculated using the maximum and minimum temperatures.

### 2.3. Field Observations: Mite Population Dynamics on Leaves

Leaf samples were collected 22 times from April (leaf buds blooming) to October (leaf fall) for both 2018 and 2019. Five trees were randomly determined, taking four twigs (approximately 15–20 cm long) per tree, each in one of the cardinal directions. Then, three leaves from each twig—one on the distal, one in the middle, and one on the proximal part of the twig—were detached and transferred to the laboratory for study (a total of 60 leaves per sampling date). Mites on the leaves were counted by direct observations under a Nikon SMZ745 stereomicroscope at a magnification of 50×. The surface area of the leaves was also calculated by means of a grid paper.

### 2.4. Field Observations: Migration to Overwintering Sites

Sticky-band traps were applied on 6 October 2018 to capture mites migrating to the overwintering sites on infested trees using the same method previously described (Figure 1C). Five trees were randomly selected for this experiment. Sticky-band traps were applied to four twigs (approximately 20–30 cm long each with about 10–12 leaves) per tree, each in one of the cardinal directions (Figure 1C). The band was placed at the twig top, just under the distal leaf, based on the principle that the apical bud contains the highest mite density (PL, data unpublished). The bands were replaced weekly with new ones and glued mites were counted under a Leitz LABORLUX S microscope at a magnification of 100×. Bands were exposed until no more mites were trapped (8 December 2018).

### 2.5. Laboratory Observations: Emergence of Overwintering Mites at Constant Temperatures and Degree-Day Calculation

To estimate the DD for the emergence of overwintering mites, the base temperature for *A. angustifoliae* development was assessed based on the growth rate method applied at seven temperatures in laboratory trials according to Bergh (1992) [17]. 

Twenty-five infested trees were randomly selected and marked with flagging tape on 1 January 2017. One hundred current year’s growth twigs (four per tree, each in one of the cardinal directions; each twig approximately 20–30 cm long with about 10–12 buds) were cut from the selected trees and stored in an incubator (3 ± 2 °C, 85 ± 5% RH) until the beginning of the experiments. The emergence rate of the overwintering mites was recorded from each twig kept under constant temperatures of 5, 7.5, 10, 12.5, 15, 17.5 and 20 ± 0.5 °C in incubators (Kavoosh Mega, Tehran, Iran) at a photoperiod of 16:8 h (L:D), without controlling RH. Experiments were performed on 12 twigs at each temperature. 

A sticky-band trap was placed on each assayed twig just under the top healthy bud (Figure 2A). Each twig was managed in order to avoid its contact with the others at the same temperature treatment to prevent mites from walking to adjacent twigs (Figure 2C). The sticky-band traps were replaced daily with new ones and the glued mites were counted as mentioned in the previous section (Figure 2B). Trap replacement and mite counting continued until no more mites were trapped for more consecutive days and at most for 40 days. The cumulative emergence rate data and the number of days to reach 50% emergence of mites were determined at each tested temperature. The analysis was then performed on the data obtained from the seven twigs per treatment.

Emergence times (days) at each constant temperature were converted to emergence rates (l/days i.e., the inverse of the average days until emergence at each temperature). A weighted and linear regression was performed using Sigmaplot 12 software, which described the relationship between emergence rate and temperature. Emergence rates at each temperature were weighted by the reciprocals of temperature to provide the best fit at lower temperatures, which are probably the most ecologically relevant. Extrapolating this linear regression line through the x-axis provided the threshold base temperature for emergence, using the x-intercept method [18]. The predicted number of DDs above the lower developmental threshold (LDT) required for median emergence in a population was determined by taking the reciprocal of the slope of this linear regression line. 

The standard error of the predicted degree day value (SE-DD) for median (50%) emergence was calculated by dividing the SE of the slope of the regression line by the slope squared [19].

In order to validate the laboratory data, DD was calculated starting from 1 January by subtracting the base temperature from the actual daily mean temperature recorded in the field, summing the result for each day to obtain the cumulative value. The results of the field observations and data collection were compared with the laboratory data.

### 2.6. Further Statistical Analysis

For the analysis of variance (ANOVA) of all traits, the combined ANOVA was performed based on a completely randomized design by means of SAS Version 9.0.3. For comparison of the means, the Duncan’s multiple range test was used at *p* ≤ 0.05.

## 3. Results

### 3.1. Field Observations: Population Dynamics of Overwintering Mites

The mean number of live mites per bud was significantly highest (75.8 mites) on 24 February 2018, whereas fewer than 10 mites per bud were found on the other sampling dates (Figure 3). 

Mite density inside the buds was much lower in the samplings of winter 2018–spring 2019 than in the previous season and followed a different dynamic pattern, with a gradual decrease of the mite density from the first—with the significantly highest value—to the last sampling dates (Figure 4).

### 3.2. Field Observations: Phenology of Mite Emergence

Mites were trapped between 13 and 31 March 2018. The peak of emergence of overwintering mites was recorded on 18 March, when bud break started (Figure 5). The cumulative emergence curve of overwintering mites (Figure 5) reached values over 50% around 17 March.

A perfect matching of the biological sequence and data was observed between the overwintering mites found inside the buds (Figure 3) and the mite emergence (Figure 5).

### 3.3. Field Observations: Mite Population Dynamics on Leaves

The first mites on leaves were found on 5 May 2018 and on 3 June 2019 (Figure 6 and Figure 7). 

The density peak in 2018 was detected from 25 August to 8 September with more than 20 mites per cm^2^ of leaf lamina (Figure 6). Predatory mites of the family Phytoseiidae (mainly *Typhlodromus* sp.) were rarely observed in 2018 sampling dates. 

The mite peak density in 2019 was recorded on 31 July but the density was very low, not exceeding three mites per cm^2^ (Figure 7). During 2019 sampling dates, a higher population density of phytoseiids (also in this case mainly composed of *Typhlodromus* sp.) was observed actively feeding on the eriophyid mites. These predatory mites appeared on 17 July, had a density peak on 31 July, and none of them were observed after 25 September (Figure 7). Therefore, the density peak of both mites was recorded on the same sampling date.

In both years, the mite population density gradually decreased in September and October down to zero, when mites migrated from the leaf surface to overwintering sites, i.e., to the buds.

The plant malformation resulting from mite activity was observed for the first time each year on 8 July 2018 and 26 June 2019, when populations of 0.98 and 1.79 mites per cm^2^ were recorded, respectively (Figure 6, Figure 7 and Figure 8D). The malformations developed along the main vein and over time along with the increase of mite density (Figure 8E–G). Leaf deformities consisted of invaginations on the leaf lamina, with the development of dense hairiness (Figure 7 and Figure 8). In autumn, infested deformed leaves turned yellow and fell off prematurely (Figure 7 and Figure 8H,I).

### 3.4. Field Observations: Migration to Overwintering Sites

The onset of the mite overwintering period in the buds in 2018 coincided with the beginning of autumn leaf color changes, on 20 October. The highest captures occurred on 3 November when the cumulative emergence curve was over 50% (Figure 9); the last captures were obtained on 24 November.

### 3.5. Laboratory Observations: Emergence of Overwintering Mites at Constant Temperatures and Degree Day Calculation

No mites were recorded on any of the twigs exposed for 40 days at 5 °C in the laboratory; data recording was stopped after that period at that temperature. The emergence rates at temperatures between 7.5 and 20 °C were plotted in order to determine the LDT and to construct a DD regression model (Figure 10). 

The linear regression model described (*p* < 0.0001, R^2^ = 0.818) the emergence rate in the selected range of temperatures. LDT was established at 4.5 °C and the emergence rate (Y) was calculated by the equation Y = 0.0094X − 0.042 (Figure 10).

At all tested temperatures, the time (days) needed for 50% emergence (T_50_ parameter) of *A. angustifoliae* was predicted by the three-parameter sigmoidal model (Figure 11, Table 1). The results indicated that the T_50_ parameter decreased from 7.5 to 20 °C. Obviously, mite emergence from the overwintering sites occurred more rapidly at higher temperatures (Table 1) and, on average, it ranged between 85.5 DD at 20 °C and 104.4 DD at 10 °C.

In order to validate the laboratory data, the average cumulative emergence of overwintering mites was ascertained in the field during 2018–2019 (Figure 12) and the emergence of 50% of the overwintering population occurred after 119 DD.

## 4. Discussion and Conclusions

The study of biology in the field and the laboratory is particularly challenging in eriophyoid mites due to their size, secluded behavior, and environmental needs, which hamper their rearing on artificial substrates for experimental trials, usually demanding extra effort from the researchers [20]. *Aceria angustifoliae* has a double ecological interest either as a biological control agent or as a pest of the RO according to its perceived status (noxious or ornamental) in different countries. The current study was focused on clarifying basic biological aspects for the first time in this species, including the evaluation of its temperature-dependent development and survival in laboratory conditions, providing relevant data for the future development of models based on degree days requirements. 

In the study of the population dynamics of *A. angustifoliae* overwintering inside the buds, an increase of mite density was observed in the middle of winter 2017–2018, different from what was detected in winter 2018–2019. In total, the density of active mites inside the buds in winter 2017–spring 2018 was higher than that in winter 2018–spring 2019. This pattern is probably related to the higher number of days with a daily mean temperature higher than the LDT (here determined) in winter 2017–spring 2018 (11 in December + 5 in January + 17 in February + 31 in March) compared to winter 2018–spring 2019 (12 in December + 1 in January + 5 in February + 25 in March) (Appendix A). In complementary evaluations conducted by the first author of this publication (unpublished), *A. angustifoliae* was observed to be able to reproduce in the buds feeding on the bud scales in winter. Therefore, the higher week/month winter temperatures may have made the overwintering mite population more active and better enabled to feed on leaf scales inside the buds, enabling them to reach a higher population density—although slowly—in winter 2017–2018 than in winter 2018–2019. Obviously, in the second half of March 2018 and in winter 2018–spring 2019, the mite density inside the buds seemed to decrease gradually. The overwintering mite population depended on the beginning of the emergence of mites from the buds as a consequence of the seasonal rising temperatures. According to our observations, the peak of mite emergence corresponded to the initial bud breaking and growth of the leaves of the RO trees. This phenological stage of the host plant might be used as an indicator for the most appropriate time for spray applications where it is considered necessary.

As expected, mite density on leaves was directly related to the seasons, as reported for other eriophyoid mites on unsprayed host plants [15], with a density peak in the middle of summer. There was a difference of about 25 days between the two population peaks in 2018 and 2019. In both years, only very light differences were observed in the population dynamics, possibly due to the presence of predators or undetected factors other than temperature. In general, the population density of mites on leaves was lower than in 2018. In 2017–2018, phytoseiid mite density was very low and they were rarely observed. In the following season, the phytoseiid population was somewhat larger and seemingly able to reduce the eriophyid mite population. As in previous studies, an effect of the presence of predators on the reduction of populations of eriophyoid mites has been suggested [21]. The predator population appeared to be directly dependent on the prey mite population, as suggested by similar patterns of fluctuation in the populations of phytoseiids and eriophyids. The current data are supported by the literature. In fact, for example, the population density of *Aculus schlechtendali* (Nalepa) on apple trees and its predator, *Anystis baccarum* L. (Prostigmata: Anystidae), followed a similar trend and correlated clearly with each other [22].

According to the present study, mites migrate towards the buds in the autumn when the leaves start to turn yellow. At this time, the leaves are older, they begin to harden, and eriophyid mites could not find suitable food [23]. In addition, the leaves will begin to fall at this time and the mites respond to autumnal changes and begin to migrate [23]. 

The DD model can be used to predict events in the life cycle of an ectothermic organism by measuring its growth rate in relation to the temperature [24,25]. Different DDs are required based on the target species, its strain, and local environmental characteristics [24,25]. Predictive models based on DD have been used for studying the influence of the temperatures on the biological parameters of eriophyoids like *Epitrimerus pyri* [26], *Aculus fockeui* (Nalepa et Trouessart) [27], *Acaphylla theae* (Watt) [28], *Phyllocoptruta oleivora* (Ashmead) [29], *Aceria guerreronis* Keifer [30], *Aculops lycopersici* (Tyron) [31], *Calepitrimerus vitis* (Nalepa) [15,32], *Aceria tosichella* Keifer [33], *Rhyncaphytoptus ficifoliae* Keifer [31], *P. avellanae* [34], and a few others.

The current study on *A. angustifoliae* allowed identification of its LDT for emergence at 4.5 °C and an emergence of 50% of the overwintering population occurred in the laboratory after 85.5–104.4 DDs. Obviously no emergence was detected at 5 °C because observations lasted only 40 days at that temperature—a duration that should not have allowed it to reach enough temperature accumulation. The assessed LDT is in accordance with that established for 90% of the other studied species [25]. The average cumulative emergence [17] of overwintering mites ascertained in 2018–2019 in the field matched quite closely the estimated DD based on the LDT (Figure 12). In fact, the emergence of 50% of the overwintering population in the field occurred after 119 DDs, which was quite close to the data established in the laboratory. Therefore, it can be said that the defined model has sufficient scientific validity for practical use in controlling this pest.

The values of DD for the emergence of *A. angustifoliae* appear to be close to those recorded by Bergh and Judd (1993) [8] for *E. pyri*, which used 6.2 °C as LDT for emergence over a period of 50 days of observations and the emergence of 50% of the overwintering mite population occurred after 62 DDs. It was closer to the emergence prediction of a 50% population for *C. vitis* deutogynes, for which the model suggested 82 DDs [31] with a LDT of 10.51 °C. To the contrary, current data on *A. angustifoliae* are quite far from those published on *P. avellanae* for which an accumulated DD of about 172 °C, with an LDT of 6 °C, was used to estimate the beginning of mite emergence [34].

The bio-ecological data of *A. angustifoliae* assessed in the current research (migration time, population dynamics, temperature threshold) might be used for planning and managing mite control. In the case of chemical application against this mite, it is needed to take into account when the mites move for intra-plant dispersal, also related to migration from old buds to new foliage and buds or from leaves to the overwintering sites. Dispersal can be considered the critical condition in which mites are more exposed on the plant surfaces and the pesticide sprays can be more successful [15]. In addition, the ecological relationships between *A. angustifoliae* and the predatory community remain to be investigated for the evaluation of species composition and distribution, their feeding habit, and their impact on the eriophyid population dynamic. This knowledge can be used to plan a strategy that could safeguard the population of this biological control agent and allow them to represent an alternative to chemical control.

## Figures and Tables

**Figure 1 insects-14-00633-f001:**
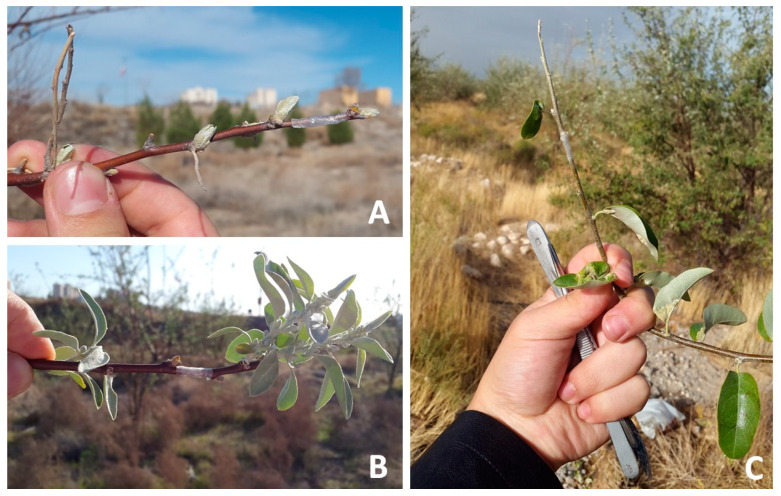
Sticky-traps mounted on twigs: (**A**) on 11 March 2018; (**B**) on 6 April 2018; (**C**) on 6 October 2018.

**Figure 2 insects-14-00633-f002:**
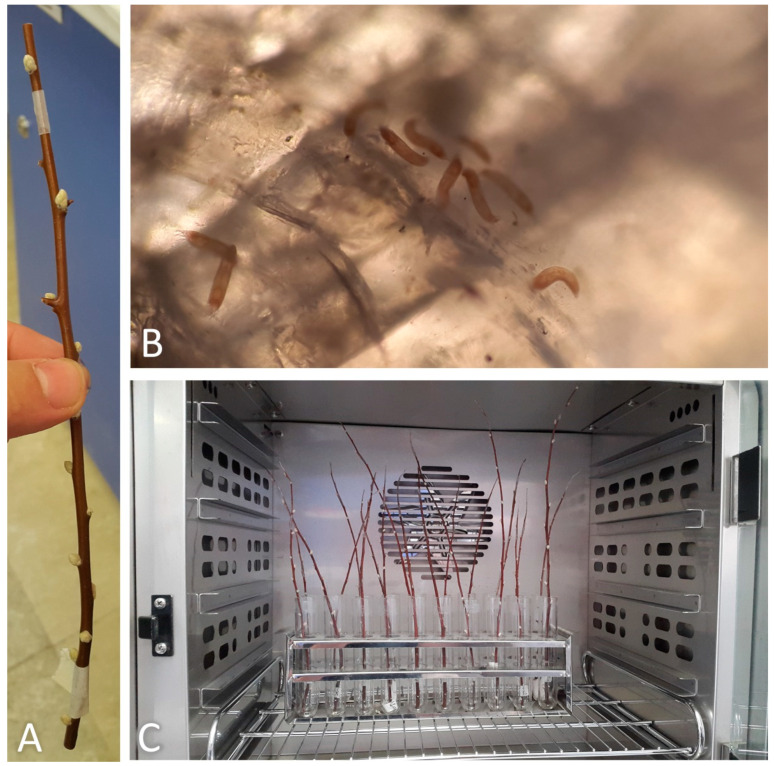
(**A**) Sticky-band trap mounted on a twig; (**B**) mites glued on a sticky-band trap observed under the microscope; (**C**) twigs inside an incubator.

**Figure 3 insects-14-00633-f003:**
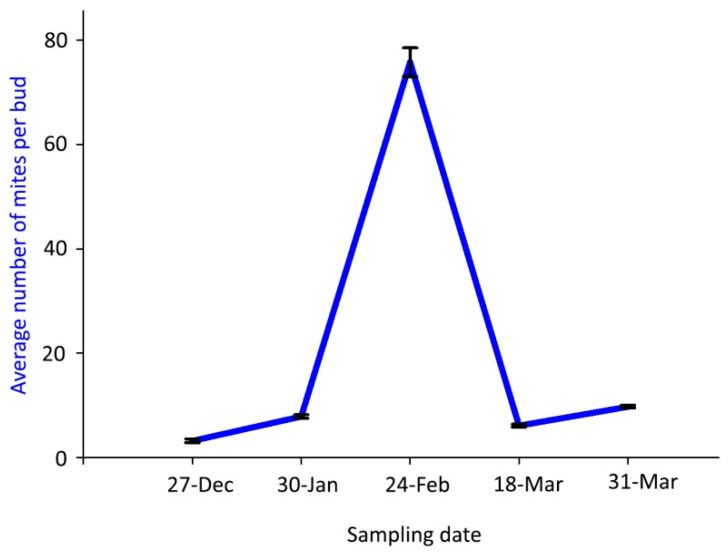
Population dynamics of *Aceria angustifoliae*: mean number of live overwintering mites inside Russian olive buds in winter 2017–spring 2018. Bars indicate the standard error range.

**Figure 4 insects-14-00633-f004:**
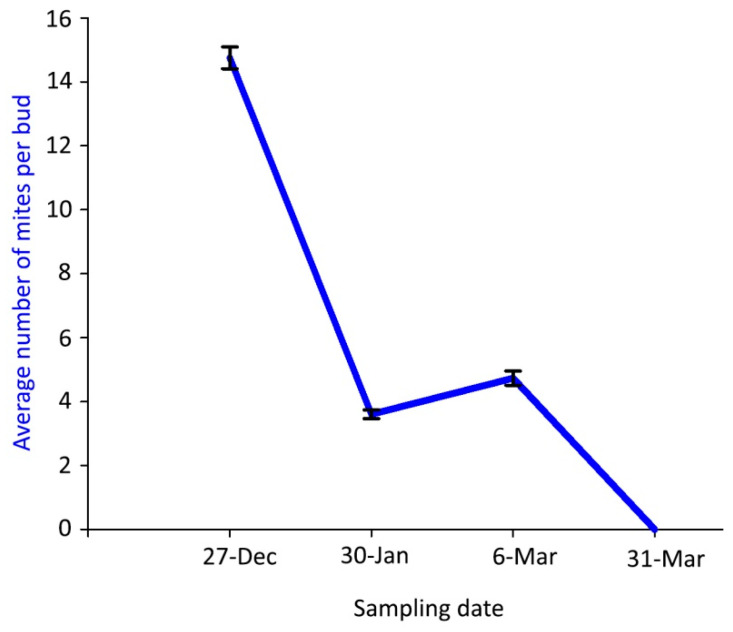
Population dynamics of *Aceria angustifoliae*: mean number of live overwintering mites inside Russian olive buds in winter 2018–spring 2019. Bars indicate the standard error range.

**Figure 5 insects-14-00633-f005:**
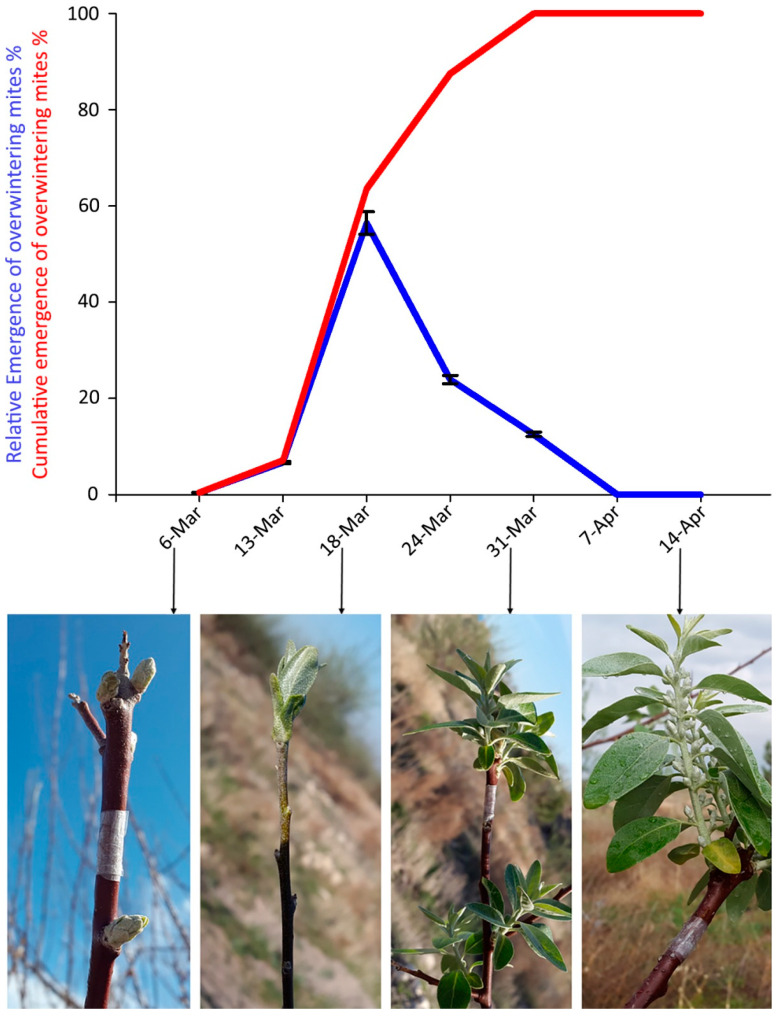
Captures of *Aceria angustifoliae* mites by the sticky-band traps mounted on Russian olive twigs during March–April 2018: (i) percentage at each sampling date on the total captured mites (blue line) and corresponding phenological stages of the infested Russian olive trees; (ii) cumulative captures (red line). Bars indicate the standard error range.

**Figure 6 insects-14-00633-f006:**
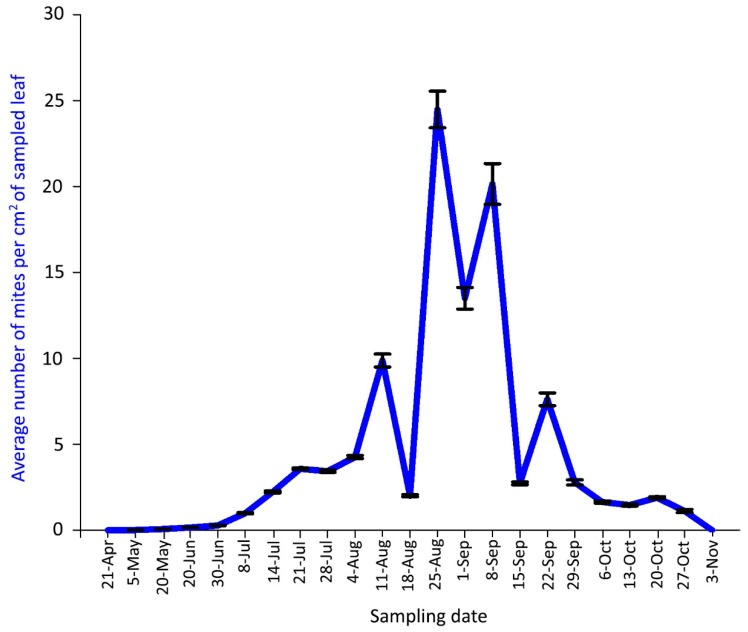
Population dynamics of *Aceria angustifoliae*: mean number of active mites on Russian olive leaves in April–October 2018. Bars indicate the standard error range.

**Figure 7 insects-14-00633-f007:**
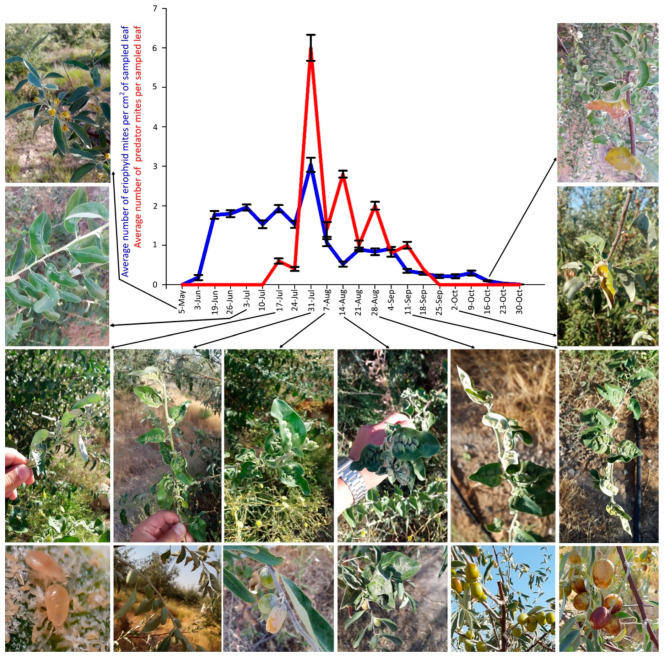
Population dynamics of *Aceria angustifoliae* and Phytoseiidae: mean number of active mites on Russian olive leaves and their correlation with phenological stages and symptoms on infested trees, in May–October 2019. Bars indicate the standard error range.

**Figure 8 insects-14-00633-f008:**
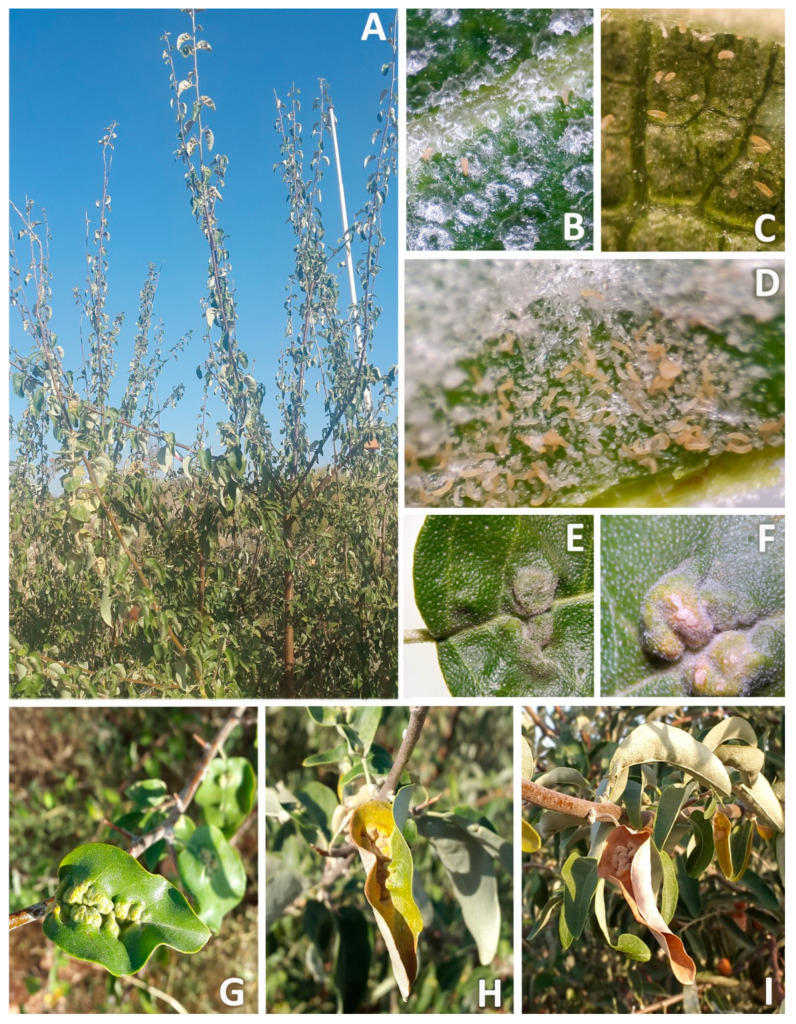
Russian olive trees infested by *Aceria angustifoliae*: (**A**) general aspect of infested trees; (**B**) leaves with low mite density early in the season; (**C**) medium mite density without any deformation on leaves; (**D**) high mite density with leaf deformations; (**E**) newly formed leaf deformations; (**F**) more advanced progress of leaf deformations; (**G**) wide leaf deformations; (**H**) yellowing of deformed leaves; (**I**) finally, drying of deformed leaves.

**Figure 9 insects-14-00633-f009:**
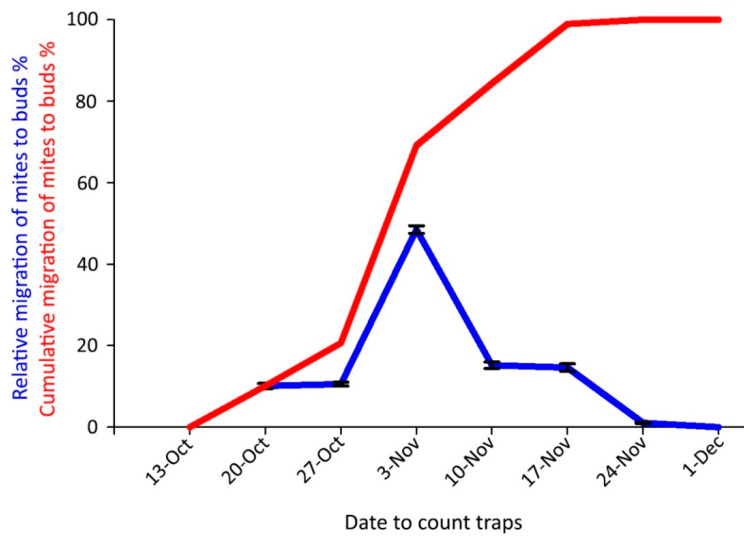
Captures of *Aceria angustifoliae* mites by sticky-band traps mounted on Russian olive twigs in October–December 2018 intercepting mites moving to the buds for overwintering: (i) percentage at each sampling date on the total captured ones (blue line); (ii) cumulative captured migrating mites (red line). Bars indicate the standard error range.

**Figure 10 insects-14-00633-f010:**
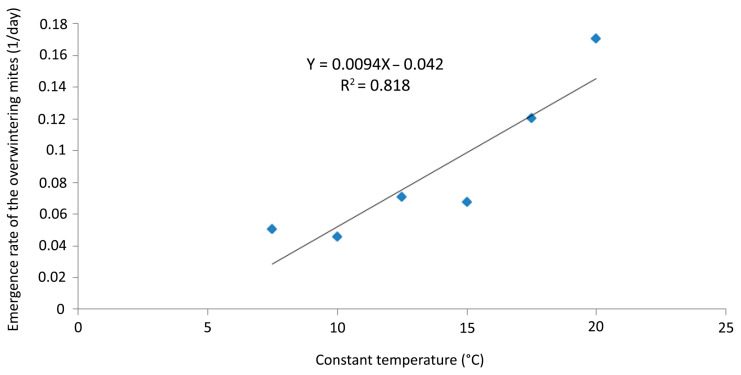
*Aceria angustifoliae***:** linear regression of the emergence rates of the overwintering mites (1/day) at six constant temperatures (7.5–20 °C) and extrapolation of the x-axis fitting line and lower developmental threshold (LDT = 4.5 °C).

**Figure 11 insects-14-00633-f011:**
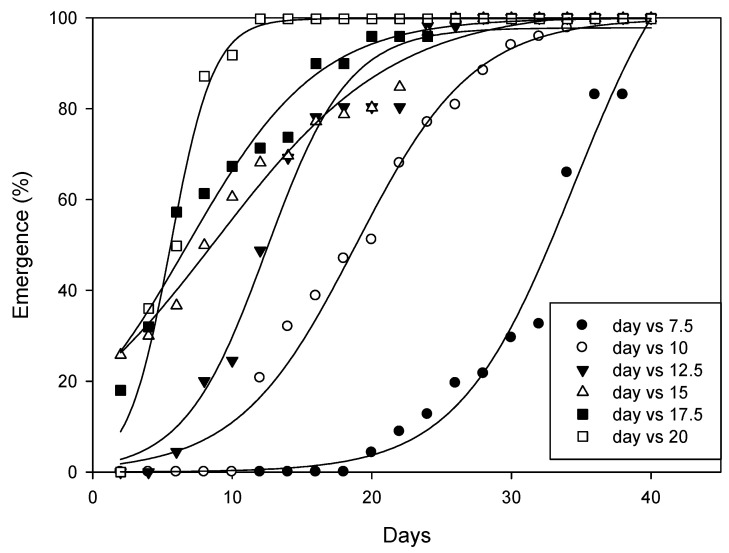
Effect of the selected temperatures on the emergence of *Aceria angustifoliae* during a 40-day period.

**Figure 12 insects-14-00633-f012:**
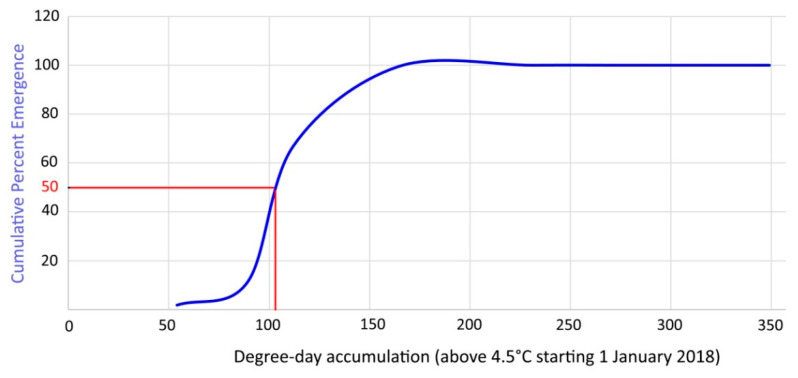
Cumulative percent emergence curve of overwintering mites based on DD above 4.5 °C starting on 1 January 2018.

**Table 1 insects-14-00633-t001:** Average values of the three-parameter sigmoid model used for calculating emergence of *A. angustifoliae* at different temperatures. E max is the maximum mite emergence percentage, T_50_ is the time (days) required for 50% mite emergence, b indicates the slope. These data were applied in Equation (1) as follows: E (%) = E max/[1 + exponential (−(T − T_50_)/b)], where E represents the cumulative mite emergence (%) at time T. Values in parentheses were standard errors (±SE).

Temperature (°C)	*E* _max_	*T_50_*	*b*	*p* Value	*R^2^*
7.5	97.42 (1.14)	34.06 (1.30)	4.13 (0.56)	<0.0001	0.98
10.0	97.05 (0.81)	18.85 (0.44)	4.20 (0.10)	<0.0001	0.98
12.5	96.65 (0.92)	12.44 (0.38)	2.95(0.33)	<0.0001	0.97
15.0	96.93 (0.76)	8.80 (0.41)	6.40 (0.48)	<0.0001	0.98
17.5	97.11 (0.93)	6. 78 (0.23)	4.71 (0.47)	<0.0001	0.96
20.0	95.93 (1.01)	5.50 (0.71)	1.54 (0.15)	<0.0001	0.98

## Data Availability

Data are contained within the article.

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
