# Peer review of "Field and Laboratory Observations on the Biology of *Aceria angustifoliae* with Emphasis on Emergence of Overwintering Mites"

_insects, 2023, doi:10.3390/insects14070633_

Round 1
Reviewer 1 Report
Lotfollahi et al. Field and laboratory observations on the biology of Aceria angustifoliae….
The manuscript provides the results from extensive field and laboratory observations on A. angustifoliae on its host Russian olive (E. angustifoliae) to delineate the timing of movement of mites out of the buds to the leaves, population fluctuations on the leaves through the season, and establishment of a temperature model to predict emergence from overwintering sites. The significance of these findings is justified for potential management of mite populations on Russian olive where it is endemic but also for potential biological control using the mites in areas where Russian olive is invasive. Extensive work was included in these studies and the findings are significant enough to warrant publication. The manuscript should be accepted; however, there are several points that need to be corrected and/or addressed before publication.
- P 2: clarify the wording of the ‘research aims’
o i. change ‘mite leaf fluctuations’ to ‘fluctuations in mite populations on the leaf’
o ii. Add ‘… and determining its developmental threshold’
o iii change ‘predict the overwintering mite emergence’ to ‘predict the emergence of mites from overwintering sites’
- Figures 3 and 4 (and others as well) need to include error bars as without some type of estimate of variance the comparative data across sampling times has really no meaning as we do not have any idea how variable the sampling method is. A statistical test to determine if any of the sampling data are significantly different from each other is also necessary.
- For the mite emergence from overwintering site data: methods need to be clarified to provide more information on how long the weekly sampling was done. Also calling this data mite emergence is a bit inaccurate. The sticky bands measured the relative mite activity (perhaps ‘spring dispersal activity’ may be a better term) on the stems as there were buds both above and below the bands. This data is still valuable as it does measure indirectly the movement from the buds. Also it is very unfortunate that the 2018-19 sampling was not done as it would have provided more evidence that you were measuring exactly what you claim when mites in buds and relative mite activity on stems correlate well as in 2017-18.
- Figures are overused: Figures 5 and 6 present the same data as does Fig 14; also Figures 10 and 11 present the same data only one figure is necessary for each of these.
- For mite on leaves data: no explanation is given why 1st mite collection a month later in 2019 than in 2018 and only info included for correlation to plant phenology stage was for 2019. Were these observed to be the same each year?
- The calculation of emergence rates is confusing. I was not clear what the emergence rate of ‘1/day’ means. This process needs to be explained better. How do the emergence rates calculated in Table 1 relate to the emergence rates used in Figure 12? Also these seem to be presented out of order as the Table 1 data are used to determine emergence rates that are determined in Figure 12 but these are presented backwards.
- With regard to the calculation of DD to 50% emergence: in the results for Figure 12 it indicates that this is 106 DD but Table 1 seems to show something different. The T50 (‘time (days) required for 50% mite emergence’) for 7.5C is 10.26 days but 10.26 days at constant 7.5C would be 3DD per day (7.5-4.5) so the total DD to T50 is about 31 accumulated DD (10.26 x 3). This also holds for the other temps in Table 1 as none of the constant temp DD accumulations to T50 are close to 106 DD. Is this a discrepancy or a misinterpretation based on confusing description?
- In the discussion the defined model shown in Table 12 is considered ‘valid’ based on 1 year of data (Fig 14). One year of data is good but certainly not enough to deem a model ‘valid’ (this is another disadvantage of not having additional 2018-19 data). Further validation will be needed in the future (but not necessary for this publication).
- In Table 12 the ‘model’ equation is NOT written correctly as seen in the paragraph above Table 12.
English quality is acceptable but could be improved and clarified in some areas with some wording changes and sentence restructuring.
Author Response
We feel to express our gratitude to both reviewers for their clear and helpful comments which surely improve the quality of the paper.
We read that both reviewers mentioned the need to improve English. We would like to inform the editor that the paper was submitted to an English revision before the original submission. This check was made by prof. James W. Amrine who is the guru of Eriophyoidea. We did not change his proofreading because he is English (American) mother tongue and because he has so huge competence with Eriophyoidea. However, we believe that the suggestions made by reviewers and our further language check can be enough valuable and the paper does not need a further professional proofreading check.
REPLY TO REVIEWER #1
COMMENT
P 2: clarify the wording of the ‘research aims’
- change ‘mite leaf fluctuations’ to ‘fluctuations in mite populations on the leaf’
- Add ‘… and determining its developmental threshold’
iii change ‘predict the overwintering mite emergence’ to ‘predict the emergence of mites from overwintering sites
REPLY
Accepted and done.
COMMENT
Figures 3 and 4 (and others as well) need to include error bars as without some type of estimate of variance the comparative data across sampling times has really no meaning as we do not have any idea how variable the sampling method is. A statistical test to determine if any of the sampling data are significantly different from each other is also necessary.
REPLY
We have considered this chance and we added a paragraph on this topic, providing further information on the figures and text. We would like to point out that this paper did not have the aim to compare periods and make a quantitative analysis. We would like to have more qualitative data at the moment, because field studies with Eriophyoidea are not so easy and these are the first data on this mite species. Standard error (SE) was reported on the graphs and ANOVA was launched. However, the SE express well the significant differences among the sampling dates and ANOVA gave also indication of significant differences among the obtained data. We decided to be smooth and we did not add further details.
COMMENT (1)
For the mite emergence from overwintering site data: methods need to be clarified to provide more information on how long the weekly sampling was done. Also calling this data mite emergence is a bit inaccurate. The sticky bands measured the relative mite activity (perhaps ‘spring dispersal activity’ may be a better term) on the stems as there were buds both above and below the bands. This data is still valuable as it does measure indirectly the movement from the buds. Also it is very unfortunate that the 2018-19 sampling was not done as it would have provided more evidence that you were measuring exactly what you claim when mites in buds and relative mite activity on stems correlate well as in 2017-18.
REPLY
About “long the weekly sampling was done”, it was reported in the text. Probably, it was not enough clear and the final sentence of the paragraph was modified.
About the inaccurateness of the terms, we do not believe we are too far from the truth, and spring dispersal activity as term is inaccurate, too, on our opinion. We mean, we expect apical buds to be the first and the strongest to develop in spring. This means that mites move from the other buds to the new stem produced by the apical bud and we do not believe that mites overwintering into the apical buds should move toward the base of the stem. Figs 1A and 1B seem to support what we are writing. In fact, in Fig. 2B we can see a richer stem (more leaves) over the sticky band. Probably, it could be continued to be used “mite emergence” explaining that dispersal of mites from other green organs cannot be excluded as well as relative mite activity. We tried to improve this part.
We were really upset about 2018-19 data, as well.
COMMENT (2)
Figures are overused: Figures 5 and 6 present the same data as does Fig 14; also Figures 10 and 11 present the same data only one figure is necessary for each of these.
REPLY
Fixed.
COMMENT (3)
For mite on leaves data: no explanation is given why 1st mite collection a month later in 2019 than in 2018 and only info included for correlation to plant phenology stage was for 2019. Were these observed to be the same each year?
REPLY
We added supplementary data about the temperature. The weather conditions were different for the two years of observations. In the second year, temperature lately started to warm and the winter climate was colder too. Supplementary data were added at the end of paper. Information are also in the Discussion paragraph.
COMMENT (4)
The calculation of emergence rates is confusing. It was not clear what the emergence rate of ‘1/day’ means. This process needs to be explained better. How do the emergence rates calculated in Table 1 relate to the emergence rates used in Figure 12? Also these seem to be presented out of order as the Table 1 data are used to determine emergence rates that are determined in Figure 12 but these are presented backwards.
REPLY
The emergence rate was calculated by 1/day and the results were presented in Fig. 12 (now 10). The "E rate" in Eq. 1 is a parameter of regression model that was estimated. The "E rate" is the slope of the regression model and is different with emergence rate (1/day). So, in Eq. 1 we used the "b" for slope instead of "E rate" in order to prevent misunderstanding (in attachment). Also in the main text the "E rate" was replaced with "b".
COMMENT (5)
With regard to the calculation of DD to 50% emergence: in the results for Figure 12 it indicates that this is 106 DD but Table 1 seems to show something different. The T50 (‘time (days)required for 50% mite emergence’) for 7.5°C is 10.26 days but 10.26 days at constant 7.5°C would be 3 DD per day (7.5-4.5) so the total DD to T50 is about 31 accumulated DD (10.26 x 3). This also holds for the other temps in Table 1 as none of the constant temp DD accumulations to T50 are close to 106 DD. Is this a discrepancy or a misinterpretation based on confusing description?
REPLY
As explained above, in Fig. 10 (based on the current sequence) we used 1/day as emergence rate in order to obtain the base thermal threshold (LDT) and the data in Table 1 was used to estimate the T50 (‘time (days) required for 50% mite emergence’). The data were revised and mistakes were found. Table and graph were replaced.
In this case we agree with the reviewer. The calculation of the emergence (%) which used in three-parameter sigmoid model (previous Figure 13) was incorrect. So the data of emergence (%) was corrected and the results of regression analysis (previous Figure 13) and the parameters of the model in Table 1 were corrected in the text. Also the revisions were done throughout the ms. Finally, Fig. 14 (currently 12) regards a real case that was used for the validation of the model.
COMMENT (6)
In the discussion the defined model shown in Table 12 is considered ‘valid’ based on 1 year of data (Fig 14). One year of data is good but certainly not enough to deem a model ‘valid’(this is another disadvantage of not having additional 2018-19 data). Further validation will be needed in the future (but not necessary for this publication).
REPLY
The reviewer was right and we were upset. However, the relevance of the study and the difficulty to work with eriophyoids pushed us to use this preliminary data. We are aware that further years of observations are needed for a stronger validation, but our research ended and we could not take much more time. It was a pity to leave these interesting data into a folder and forget them. For this reason we thought to publish them taking in count the procedures and protocols we applied which can be useful for other researchers and other eriophyoid mites. However, the comment induced us to revise the crude data and check if something was lost or omitted. We found that something was needed to be improved. We started DD count on 1 January instead of 6 March. We noted that DD was a bit higher at lower temperature than DD at higher temperature. The difference is not too big, but it is quite interesting from a biological point of view and need more attention in future research. We pointed out this observation.
COMMENT (7)
In figure 12 the ‘model’ equation is NOT written correctly as seen in the paragraph above figure 12.
REPLY
It was fixed. The corrected figure 12 is in attachment.
FURTHER COMMENT ON THE TEXT FILE
Instead of finishing the "Results" section with a table and a figure, I suggest at least a paragraph be introduced with some further comments about their contents.
REPLY
We do not believe that further comments should be added. Information and data were previously provided and any further sentence will not add any useful information.
Reviewer 2 Report
This is an interesting manuscript, that should be published. I have quite a few suggestions, shown in the attached file. I congratulate the authors for their work, and in revising it, my sole interest is to collaborate with the authors in their meticulous work.
I have indicated in the attachment all my suggestions.

The manuscript is well written. However, because of the complexity of the theme of the work and because of the many different aspects delt with, in my view the manuscript needs some improvements in que quality of the English Language. I tried to colaborate in this sense, but in my understanding, more is necessary.
Author Response
We feel to express our gratitude to both reviewers for their clear and helpful comments which surely improve the quality of the paper.
We read that both reviewers mentioned the need to improve English. We would like to inform the editor that the paper was submitted to an English revision before the original submission. This check was made by prof. James W. Amrine who is the guru of Eriophyoidea. We did not change his proofreading because he is English (American) mother tongue and because he has so huge competence with Eriophyoidea. However, we believe that the suggestions made by reviewers and our further language check can be enough valuable and the paper does not need a further professional proofreading check.
REPLY TO REVIEWER #2
We have accepted all suggestions reported on the attached file as track option and comments. We cannot add any detailed response because all of them were considered pertinent.
Round 2
Reviewer 1 Report
The authors have addressed nearly all the issues that were raised in the review. The research on these mites would be very difficult to perform and the document represents some good baseline information on this mite. The manuscript should be accepted for publication and will provide an important contribution on this mite.
One point I encourage the authors to consider relates to the term '1/day'. This term is confusing but I assume it means the inverse of the average days until emergence at each temperature. Clarity on this would help the manuscript.
The paper would benefit from some editing of the English, particularly on the numerous edits that have been added to the paper.
Author Response
Thanks for your comments. We added the explanation in Materials and methods of 1/day as suggested.
Reviewer 2 Report
The quality of the paper improved considerably in comparison with the first version. I have just a few (but important) suggestions now. Please see attachment. One of what I consider to be most important suggestions refer to the convenience to give some details about the methodology referring to the laboratory evaluation of the effect of temperature (type of equipments, precision of average temperatures, humidity and light conditions. Another important aspect refer to a presentation of a figure as part of "Discussion". I would strongly recommend to move it to "Result", despite the fact that it is based on the analysis of the results. Of course, the discussion of that relation shown in that figure should be cited in "Discussion". If possible at all, please mention at least the genus to which the dominat phytoseiid species belonged. That would be important. In case these mites were not mounted for identification, authors could cite that in discussion and at the same time cite species found on the same host in that geogrphic area previously by other authors (if available). I congratulate the authors for the beautiful work!

Author Response
Great to have your approval. Your further comments have been all accepeted and considered in the last draft we are submitting. Just one comment regard the SE in fig. 3. The range is too small. It was reported but it is not so evident.